# A chalcogenide-cluster-based semiconducting nanotube array with oriented photoconductive behavior

Jiaqi Tang [1,2,5], Xiang Wang [2,5], Jiaxu Zhang[2,5], Jing Wang[3], Wanjian Yin [3], Dong-Sheng Li[4] & Tao Wu[1✉]

The interesting physical and chemical properties of carbon nanotubes (CNTs) have prompted the search for diverse inorganic nanotubes with different compositions to expand the number of available nanotechnology applications. Among these materials, crystalline inorganic nanotubes with well-defined structures and uniform sizes are suitable for understanding structure–activity relationships. However, their preparation comes with large synthetic challenges owing to their inherent complexity. Herein, we report the example of a crystalline nanotube array based on a supertetrahedral chalcogenide cluster, $K_3[K(Cu_2Ge_3Se_9)(H_2O)]$ (1). To the best of our knowledge, this nanotube array possesses the largest diameter of crystalline inorganic nanotubes reported to date and exhibits an excellent structure-dependent electric conductivity and an oriented photoconductive behavior. This work represents a significant breakthrough both in terms of the structure of cluster-based metal chalcogenides and in the conductivity of crystalline nanotube arrays (i.e., an enhancement of ~4 orders of magnitude).

[1] College of Chemistry and Materials Science, Guangdong Provincial Key Laboratory of Functional Supramolecular Coordination Materials and Applications, Jinan University, Guangzhou 510632, China. [2] College of Chemistry, Chemical Engineering and Materials Science, Soochow University, Suzhou 215123, China. [3] College of Energy, Soochow University, Suzhou 215006, China. [4] College of Materials and Chemical Engineering, Hubei Provincial Collaborative Innovation Centre for New Energy Microgrid, Key Laboratory of Inorganic Nonmetallic Crystalline and Energy Conversion Materials, China Three Gorges University, Yichang 443002, China. [5]These authors contributed equally: Jiaqi Tang, Xiang Wang, Jiaxu Zhang. ✉email: wutao@jnu.edu.cn

Over the past three decades, one-dimensional (1D) hollow tubular nanomaterials have long been the focus of considerable interest in the physics, chemistry, and materials communities because of their unique physical and chemical properties, as well as their diverse potential applications since the discovery of carbon nanotubes (CNTs) by Iijima in 1991[1–8]. To date, several multifarious nanotubular architectures with pure inorganic or inorganic–organic hybrid or pure organic components as the nanotube skeleton have been synthesized and applied in various fields, such as catalysis[9,10], molecular capillaries[11], energy storage[12,13], and biological models[14,15]. Among the various nanotubular materials, robust inorganic nanotubes with high thermal and chemical stabilities are considered promising for a wide range of interesting potential properties and applications, and therefore, these materials have attracted a great deal of scientific attention[2,7].

Generally, inorganic nanotubes are formed by rolling up exfoliated two-dimensional (2D) flat sheets of corresponding lamellar materials under special non-equilibrium conditions (chemical vapor deposition, flash evaporation, and discharge evaporation)[16,17] or constructed through a bottom-up approach from basic inorganic elements[6,7,16–25]. The nanotubes fabricated via the rolling-up approach are usually in the form of fully or partially disordered arrays with large size distributions (Fig. 1a), which may require additional purification steps to avoid affecting their applications[6,7,19]. In contrast, high-quality nanotubes, synthesized using a bottom-up strategy and possessing an atomically precise structure and uniform size, are highly desirable for understanding structure–property relationships and future applications[7,20]. Moreover, this type of nanotube is often negatively charged and can be assembled into crystal arrays by ionic or other weak interactions (Fig. 1b), thereby providing a platform for studying ion transportation and ionic conduction within or outside the nanotubes;[6,7,17] this is important in the context of nanoelectronics and biotechnology[26]. However, only a small number of well-defined crystalline inorganic nanotube arrays have been reported owing to synthetic challenges, such as a poor design due to the inherent complexities of synthetic processes for pure inorganic materials[16], thereby contrasting with the case of rich inorganic–organic hybrid tubular structures (e.g., metal–organic nanotubes)[3,4,11,26]. In addition, the majority of crystalline inorganic nanotube arrays have been constructed using metal oxides, such as PTC-118 ({(EMIm)$_3$[(H$_2$O)⊂Ti$_6$O$_6$($\mu_2$–OH)$_3$(SO$_4$)$_6$]}$_n$), as

recently reported by Zhang et al.[7]. To date, a limited number of chalcogenide-based crystalline tubular compounds have been documented, and their photo-/electroconductivity properties remain unexplored;[16,17,20] however, they are particularly desirable for applications in nanoelectronics and optoelectronics as semiconducting chalcogenides could be advantageous in opto-/electronic property compared with insulating oxides thanks to the lower electronegativities of S/Se/Te compared with that of O[27]. The development of chalcogenide-based crystalline nanotube arrays and subsequent study of their potential photo-/electroconductivity properties and corresponding structure-activity relationships for high-performance photoelectric conversion devices are therefore of particular interest. In addition, compared with atomic-layered single-wall nanotubes (Fig. 1c), the crystalline inorganic nanotube arrays assembled by clusters could possess greater numbers of exposed sites or external surfaces due to the significantly more rugged surface constructed by protruding clusters (Fig. 1d); this could also result in interesting properties.

Thus, we herein report the preparation of a supertetrahedral chalcogenide-cluster-based compound {K$_3$[K(Cu$_2$Ge$_3$Se$_9$)(H$_2$O)]} (1), featuring a 1D nanotubular structure, and examination of its oriented photoconductive property. We expect that this structure will be distinct from the traditional 0D discrete clusters, 1D chains, 2D layers, and 3D frameworks constructed by supertetrahedral chalcogenide clusters during the past 50 years[28–30] and that it will constitute another significant breakthrough in the field of supertetrahedral clusters since the emergence of the supertetrahedral [Na$_4$Ge$_4$S$_{10}$] T2 cluster ("T" denotes tetrahedral, two denotes the number of Ge sites along the tetrahedron edge) in 1971[28,31–35].

## Results

**Crystal structure**. Red rod crystals of 1 (Supplementary Fig. 1) were synthesized via the solvothermal method (see the Experimental section in the Supporting Information for details). Single-crystal X-ray diffraction (SCXRD) analysis revealed that 1 crystallized in a highly symmetrical trigonal system with an *R*-3 space group (Supplementary Table 1), and exhibited a unique nanotubular structure (Fig. 2). The asymmetric unit of 1 contains 19 crystallographically independent sites, comprising two Cu, three Ge, and nine Se, in addition to one water guest molecule, four potassium counter-cations (Supplementary Fig. 2). In addition, the valence state of Cu was confirmed to be monovalent by X-ray photoelectron spectroscopy (XPS)

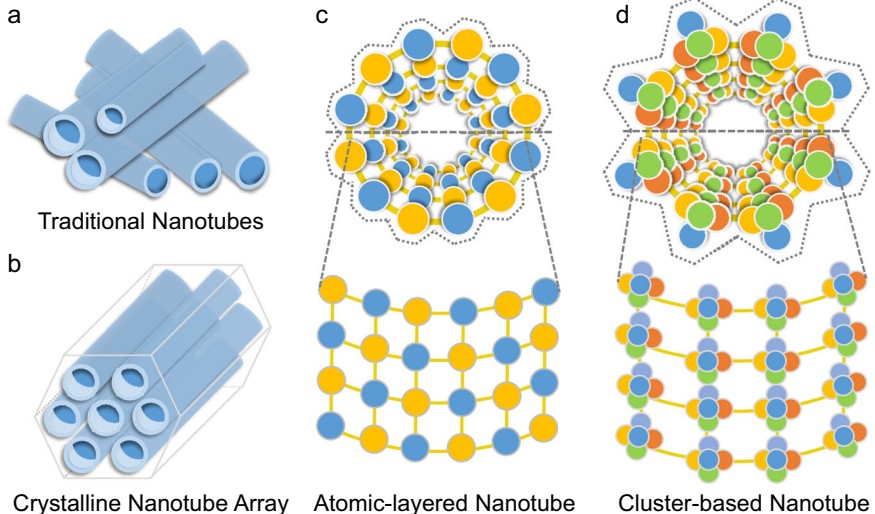

**Fig. 1 Two known types of nanotubes.** Comparisons of (**a**) traditional nanotubes and (**b**) a crystalline nanotube array. Comparison of (**c**) an atomic-layered nanotube and (**d**) a cluster-based nanotube.

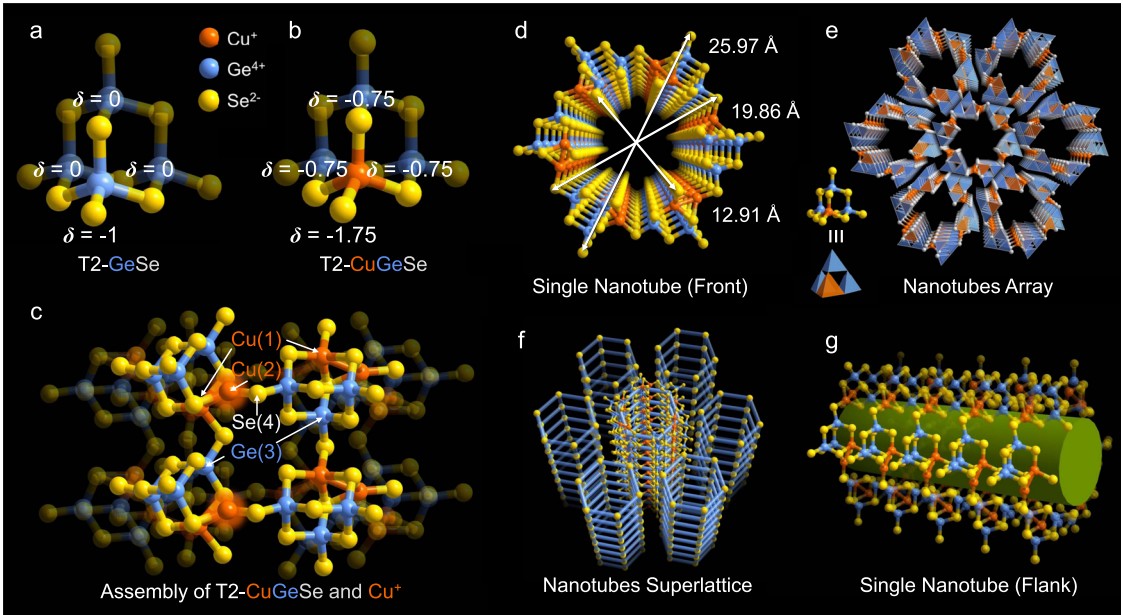

**Fig. 2 Structure of 1. a** Traditional homometallic supertetrahedral T2-GeSe cluster. **b** Heterometallic supertetrahedral T2-CuGeSe cluster ($\delta$ represents the theoretical residual charge of the Se atoms surrounding the corner Ge or Cu sites). **c** The connection mode of the T2-CuGeSe cluster in 1. **d** As-formed single 1D nanotube of 1 viewed along the *c*-axis. **e** Packing diagram of 1 viewed along the *c*-axis. **f** Pillared stacking of the connected wheel clusters in the axial direction. **g** Flank view of a single nanotube. Colors: orange, Cu; blue, Ge; yellow, Se.

(Supplementary Fig. 3), while the phase purity was verified using powder X-ray diffraction (PXRD) (Supplementary Fig. 4). Through the combination of the results of SCXRD, energy-dispersive spectroscopy (Supplementary Fig. 5), thermogravimetric analysis (TGA) (Supplementary Fig. 6), and elemental analysis, the empirical formula of 1 was determined to be $K_4Cu_2Ge_3Se_9(H_2O)$.

As shown in Fig. 2, the primary building unit (PBU) of 1 is a heterometallic supertetrahedral T2-CuGeSe cluster, which can be viewed as a homometallic supertetrahedral T2-GeSe cluster with one of the four corner germanium sites occupied by copper (Cu (1) in Supplementary Fig. 2). The $[Cu(1)Se_4]$ tetrahedron unit is slightly distorted in comparison with the $[GeSe_4]$ tetrahedron unit in the homometallic supertetrahedral T2-GeSe cluster (Fig. 2a, b), likely due to the mismatch of the local charge caused by the lower valence state of Cu compared with that of Ge. In the traditional view, these six staggered PBUs are connected end-to-end by six Cu ions to form a giant hexagonal wheel-shaped cluster ($[Cu_6(CuGe_3Se_{10})_6]$) with the $C_{3i}$ point group (Fig. 2c and Supplementary Fig. 7), which can serve as the secondary building units (SBUs) for further assembly. More specifically, in this wheel-shaped cluster, every single independent Cu(2) atom is located close to the Cu(1) site and interlinked two adjacent T2-CuGeSe clusters via three Cu–Se bonds (Fig. 2c). Among the three Cu–Se bonds, two originate from the bonding of the Cu(2) atom with two edges Se atoms next to the Cu(1) site of one T2 cluster, while the remaining Cu–Se bond comes from the bonding of the Cu(2) atom with the corner $Se(4)^{2-}$ of the other T2 cluster. Furthermore, the three Cu–Se bonds are coplanar with unequal length, and the corresponding Se–Cu–Se bond angles are also diverse from one another (Supplementary Fig. 8). It should be noted that the planar trigonal coordination mode of Cu between two supertetrahedral clusters is rare[36], in contrast to the single linearly-coordinated $Cu^+$ species between clusters[37–39] and the tetrahedrally-coordinated $Cu^+$ residing in the cluster. According to Pauling's electrostatic valence rule, the theoretical residual charges of the $Se^{2-}$ ions coordinated with the $Cu^+$ ions in an isolated T2-CuGeSe cluster, all increased by $-0.75$ when replacing one $Ge^{4+}$ cation from an isolated T2-GeSe cluster with

one low-valent $Cu^+$ cation (Fig. 2a, b), thereby resulting in a serious mismatch of local charges. To address this issue, a single low-valent $Cu^+$ cation was introduced and triangularly coordinated with three $Se^{2-}$ ions between two T2 clusters to maximally balance the excess local negative charge. Meanwhile, accompanying the looser geometrical demand of selenide compared with sulfide, an exceptional heterometallic supertetrahedral T2 cluster-based wheel-shaped ring was formed, which is in sharp contrast to the 3D frameworks constructed by homometallic T2-GeS sulfide clusters with linearly-, trigonally- or tetrahedrally-coordinated low-valent transition metal ions (such as $Cu^+$, $Ag^+$, and $Mn^{2+}$)[36–41]. To further lessen the excess high negative charge at the terminal Se site that is linked with the Cu(1), and facilitate the global charge balance, these large wheels are connected end-to-end through sharing of the terminal $Se^{2-}$ ion close to the Cu(1) site with the $Ge(3)^{4+}$ of the other T2 cluster, ultimately forming a supertetrahedral chalcogenide-cluster-based infinite nanotube ($[Cu_6(CuGe_3Se_{10})_6]_n$) along the *c* direction, with an outer diameter of 25.97 Å and an inner opening window of $12.91 \times 19.86$ Å (Fig. 2c, d). To the best of our knowledge, this structure represents the largest example of a crystalline inorganic nanotube to date (Supplementary Table 2). Moreover, the wall of the nanotube was composed of 16-membered ring (16 MR) windows formed by four T2-CuGeSe clusters with pore sizes of $3.59 \times 6.47$ Å (Supplementary Fig. 9). Because 1 cannot be dispersed in common solvents, an ultrathin section sample of 1 was observed by high-resolution transmission electron microscopy (HRETM) (Supplementary Fig. 10), revealing the high crystallinity of 1 with distinct interplanar lattice fringes of 0.78 nm, which is consistent with the observation of an XRD peak at $2\theta = 11.7°$ (Supplementary Fig. 4) according to the Bragg's equation in Supplementary Methods.

Furthermore, the negative charges on the skeleton of the nanotube are balanced by pure $K^+$ ions located at the gaps between the nanotubes and by hydrated $K^+$ ions filling in the channel. These $K^+$ ions, along with guest water molecules residing inside and outside of the nanotube, play a key role in constructing and stabilizing the Nanotubes and promote their

further packing into a highly ordered honeycomb-like hexagonal symmetrical array (Fig. 2e) via complex weak interactions such as hydrogen bonding and electrostatic interactions (Supplementary Fig. 11)[6,7,20]. Thus, control experiments demonstrated that $K_2S$ is indispensable for the formation of 1. Fig. 1f also displays the pillared stacking of the wheel clusters in the axial direction and the assembly of nanotubes to form a 1D tubular superlattice. The nanotube can also be viewed in a different way, where six 1D chains, formed by the end-to-end linkage of T2-CuGeSe clusters through sharing corner $Se^{2-}$ ions coordinated with Cu(1) and Ge (3) atoms, bind alternately with six $Cu^+$ ions in the same manner as above to form nanotubes (Fig. 2c, g, and Supplementary Fig. 12). This assembly mode is supported by the observation of 2, a 1D chain structure based on T2-CdGeSe clusters (Supplementary Fig. 13), which forms upon replacing the copper salt with a cadmium salt during preparation. Compound 2 was comprehensively characterized (Supplementary Figs. 14–18 and Supplementary Table 3), and upon comparison with the structure of 1, $Cd^{2+}$ was found to occupy the $Cu^+$ site of the T2-CuGeSe cluster in 2, resulting in decreased theoretical residual charges from the surrounding Se atoms, which correspondingly reduces the further bonding capability of the $Se^{2-}$ ions on the edges of the cluster toward other metal ions, thereby leading to the formation of 1D chains rather than 1D tubules. We, therefore, speculated that the dissimilar ionic radii and coordination modes between $Cd^{2+}$ and $Cu^+$ may also contribute to such differences.

**Electrical conductivity measurement**. The optical indirect bandgap of 1 was calculated to be 1.03 eV from the transformed solid-state UV–Vis diffuse reflectance spectrum (Supplementary Fig. 19). This value was considered relatively narrow and largely red-shifted by 0.63 eV compared with the corresponding value of 2 (i.e., 1.66 eV), thereby indicating the superior conductivity of 1[42]. This was confirmed by electrical conductivity measurements on a single crystal of 1 through a direct-current two-terminal method (Fig. 3a, b).

As shown in Fig. 3c, the electric conductivity of 1 was determined to be $7.60 \times 10^{-6}$ S cm$^{-1}$ at 40 °C along the $c$ axis and was positively related to temperature, exhibiting typical semiconductive characteristics. The corresponding activation energy ($E_a$) was calculated to be 0.52 eV (Fig. 3d). The electrical conductivity of 1 was found to be ~10,000-times higher than those of other crystalline nanotube arrays, and among one of the highest values for crystalline semiconductor materials containing copper or/and chalcogenide elements (Supplementary Table 4)[6,7,43–46]. In addition, the repeatability of the conductivity of 1 is demonstrated on five individual devices, and the results showed that the electrical conductivities of five devices are in a narrow range (Supplementary Table 5). On the other hand, the photoconductivity of 1 was investigated. As shown in Figs. 3e and 1 exhibit a rapid wavelength-dependent response upon illumination with 400–700 nm light, without any apparent attenuation during the on/off switching cycles, thereby indicating the efficient separation of photogenerated charge carriers[47]. The responsivity ($R_\lambda$), detectivity ($D^*$), and external quantum efficiency (EQE) at different wavelengths are summarized in Supplementary Table 6. Interestingly, the largest value of $R_\lambda$ was achieved at 600 nm (Fig. 3f), which is inconsistent with the maximum absorption in UV–Vis spectrum (Supplementary Fig. 19a), thereby suggesting a temporary unclear process that enhanced the photocurrent at longer wavelengths[47]. Moreover, the $R_\lambda$ and $D^*$ values of 1 increased as the light intensity decreased (Supplementary Fig. 20). Thus, with its outstanding conductivity, fast turn-over response, and good reproducibility, 1 displays potential for use in optoelectrical applications[47]. In the context of 2, the conductivity was determined to be only $9.1 \times 10^{-9}$ S cm$^{-1}$ at 40 °C, with an $E_a$

of 0.64 eV (Supplementary Fig. 21), while its photoconductivity performance was also significantly poorer than that of 1 (Supplementary Fig. 22). The enormous conductivity disparity between 1 and 2 was mainly attributed to their different structures and compositions. As 1 possesses more complex unidirectional connectivity than 2, it is helpful for the transport of electrons in 1. In addition, copper is much more conducive to electron transport than cadmium, which may also contribute to much-improved conductivity of 1 than 2. Thus, we speculated that the substitution of Cd with Cu in the 1D T2-CdGeSe chain may largely improve the intrinsic conductivity due to the superior conductivity of Cu compared with Cd. Combined with the narrow optical bandgap, a good oriented photoconductive behavior can be observed in 1.

## Discussion

To gain deep insight into the intrinsic electronic properties of 1, density functional theory (DFT) calculations on the band structure and the projected density of states (PDOS) were performed, whereby 1 was found to exhibit a quasi-direct bandgap of 0.92 eV at the gamma point (left of Fig. 4a) due to the small bandgap difference between direct and indirect gap (0.01 eV)[48], consistent with the experimental value.

Compared with the almost flat band lines close to the valence band maximum (VBM) along the whole Brillouin zone, which are mainly dominated by the Cu $d$ orbital and the Se $p$ orbital (right of Fig. 4a and Supplementary Fig. 23), the bands near the conduction band minimum (CBM), which are contributed primarily by the Ge $s$ orbital and the Se $p$ orbital, along with negligible contributions from the Cu $d$ orbital (right of Fig. 4a and Supplementary Fig. 23), show a significantly steep dispersion with an energy difference of ~ 0.71 eV (0.63 eV) along with the $\Gamma \rightarrow A$ ($K \rightarrow H$ and $M \rightarrow L$) directions in reciprocal space, corresponding to the tubular direction (or $c$ axis) in real space, while for other paths with high symmetry points, the dispersion widths are small (maximum energy difference < 0.1 eV). The relatively large dispersion strength of the energy bands indicates the facile transport of charge carriers along the $c$ direction[42,49], which is of paramount importance to photoconductive devices. Moreover, the narrow bandwidth and flat band lines near the VBM were attributed to the relatively larger localization of the Cu $d$ orbital compared with the Ge $s$ orbital. The other elements (K, O, and H atoms) do not contribute to the electronic band edges. In addition, according to the charge density distributions of the VBM and the CBM (Fig. 4b–e), the CBM was determined to be mainly localized on the Se atoms and the Ge(2) and Ge(3) atoms, while Ge(1) atoms make no contribution. Therefore, combined with the above analysis and the features of the crystal structure, we deduced that the excellent conductivity of 1 may be attributed to the more facile oriented transport of electrons in the tubular direction (or along the $c$ axis), in addition to obstructed carrier transport in the $ab$ plane perpendicular to the tubular direction.

In summary, we report a supertetrahedral chalcogenide cluster-based crystalline inorganic nanotube array, representing an important step toward nanotube materials. The fine electrical conductivity oriented photoconductive property, and well-defined structure of 1 render it a fascinating structural model in the optoelectronic and electronic fields. In addition, the precise potassium ions located around or within the nanotubes introduce a platform for the further study of ion transport. Finally, research exploring the syntheses of supertetrahedral chalcogenide cluster-based nanotubes with attractive functions and properties, such as ion exchange and sensing, are currently underway, and the results will be presented in the future work.

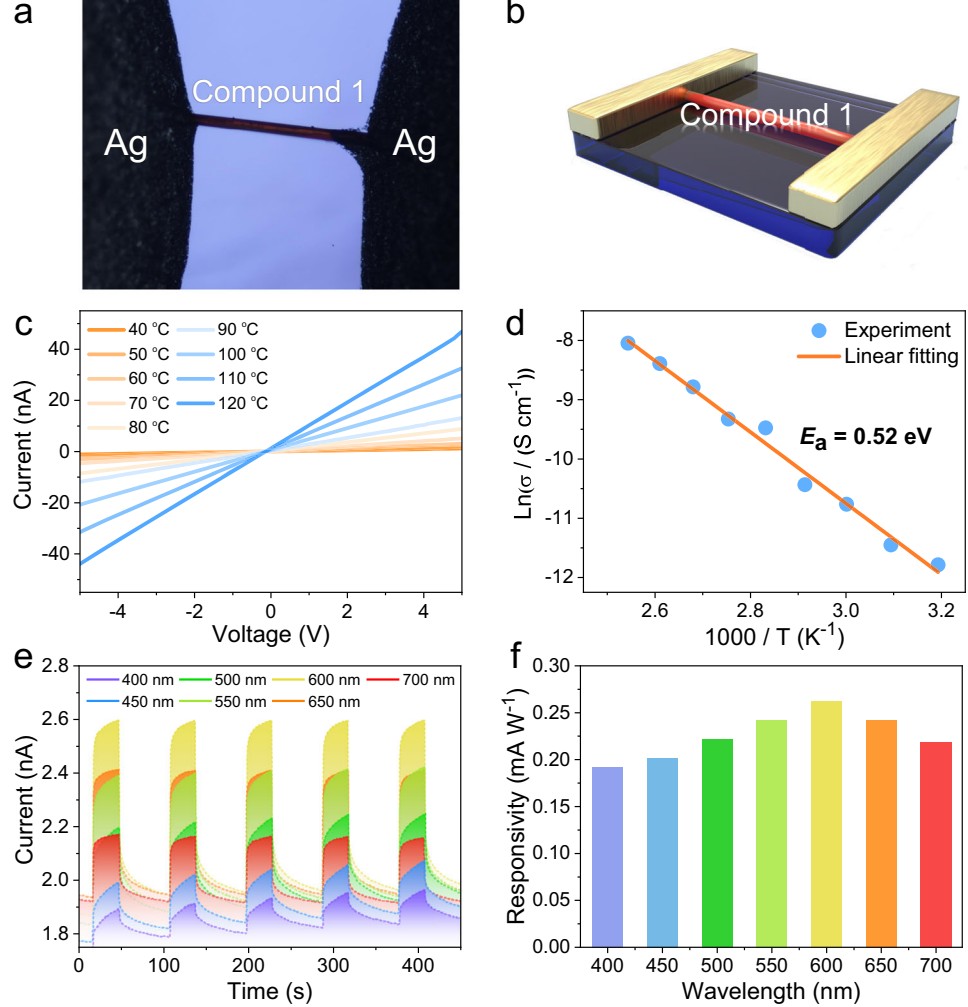

**Fig. 3 Electrical conductivity and photoconductivity studies of 1. a** Optical photographic image, and (**b**) a schematic diagram of the single crystal of 1 used for electrical measurements. **c** Temperature-dependent *I–V* curves, and (**d**) corresponding Arrhenius plots for 1. **e** Time-dependent photocurrent response curves, and (**f**) responsivities of 1 under illumination at different light wavelengths. Source data for panels (**c**)–(**e**) are provided as a Source Data file.

## Methods

**Materials**. All chemicals were analytical grade and purchased from Sinopharm Chemical Reagent Co., Ltd. Germanium dioxide (GeO$_2$, 99.99%), copper acetate hydrate (Cu(Ac)$_2$·H$_2$O, 99%), cadmium acetate dihydrate (Cd(Ac)$_2$·2H$_2$O, ≥ 98%), selenium powder (Se, ≥99.99%), deionized water (H$_2$O), (±)-2-amino-1-butanol (2-AB, 98%), 1,8-diazabicyclo[5.4.0]undec-7-ene (DBU, 99%), potassium sulfide (K$_2$S, 99%) and were all used without any further purification.

**Synthesis of compound 1**. Germanium dioxide (54 mg, 0.52 mmol), copper acetate hydrate (50 mg, 0.25 mmol), selenium powder (240 mg, 3.04 mmol), potassium sulfide (90 mg, 0.82 mmol), DBU (2.00 mL), (±)-2-amino-1-butanol (1.00 mL) and deionized water (1.00 mL) were mixed in a 25 mL Teflon-lining stainless steel and stirred for 30 min, then heated to 180 °C for 9 days. After cooled down to room temperature, all the products in the autoclave are transferred to a 10 mL glass vial, and the upper mother liquor is abandoned. Then ca. 2–4 mL ethylenediamine is added to the glass vial with sonication (40 kHz, 240 W) for ca.1 min. After standing for 1 min, the upper liquid is abandoned. Repeat the above steps with ethylenediamine until the upper liquid becomes clear with a lighter color. Then repeat the above steps with ethonal 2–3 times to remove ethylenediamine. Finally, a small amount of clean and pure red rod crystals could be obtained after filtration (yield: <1%, based on Cu element). Note, adding a small amount of trimesic acid (~10 mg) in the synthesis process could increase the crystal yield. To avoid being oxidized and the decomposition of structure, 1 is usually stored in the glove box filled with N$_2$ for further use.

**Synthesis of compound 2**. Germanium dioxide (104 mg, 0.99 mmol), cadmium acetate dihydrate (72 mg, 0.27 mmol), selenium powder (180 mg, 2.28 mmol), potassium sulfide (90 mg, 0.82 mmol), DBU (2.00 mL), (±)-2-amino-1-butanol (1.00 mL) and deionized water (1.00 mL) were mixed in a 25 mL Teflon-lining stainless steel and

stirred for 30 min, then heated to 180 °C for 9 days. After cooling down to room temperature, a small amount of yellow rod crystals could be obtained by means of the same treatment used in 1. To avoid being oxidized and the decomposition of structure, 2 is usually stored in the glove box filled with N$_2$ for further use.

**Electrical conductivity measurement**. Before preparing for electrical contact, the cylindrical crystals were covered by silver paste and connected to the semiconductor analysis system (4200SCS, Keithley) by gold wires on the insulating sapphire substrate. The lengths and widths of the column-shaped crystals on the substrate were estimated by means of a microscope based on the width of the gold wire (diameter: 50 μm). Electrical conductivity ($\sigma$) is obtained by fitting the linear region of the current–voltage curves according to Ohm's law. The activation energies reported herein were calculated by the Arrhenius Eq. (1) as

$$\ln k = \ln A - \frac{Ea}{RT} \qquad (1)$$

The temperature-dependent *I–V* curve measurements for the single crystal of 1 and 2 were performed on KEITHLEY4200-SCS by means of a direct current two-terminal method. The temperature is controlled by a digital automatic temperature control oven (STIK BAO-35A) and the measurements are performed in the range of 20–120 °C. Each measurement was performed on several individual single crystals of compounds 1 and 2.

**Photodetector fabrication and measurement**. The metal chalcogenide-based photodetection device was fabricated by placing the single crystal between two gold electrodes glued by electrically conductive silver paste. For photodetection characterizations, the device was perpendicularly illuminated by monochromatic light with different wavelengths in a vacuum. The photocurrent was recorded through a semiconductor characterization system (4200-SCS, Keithley).

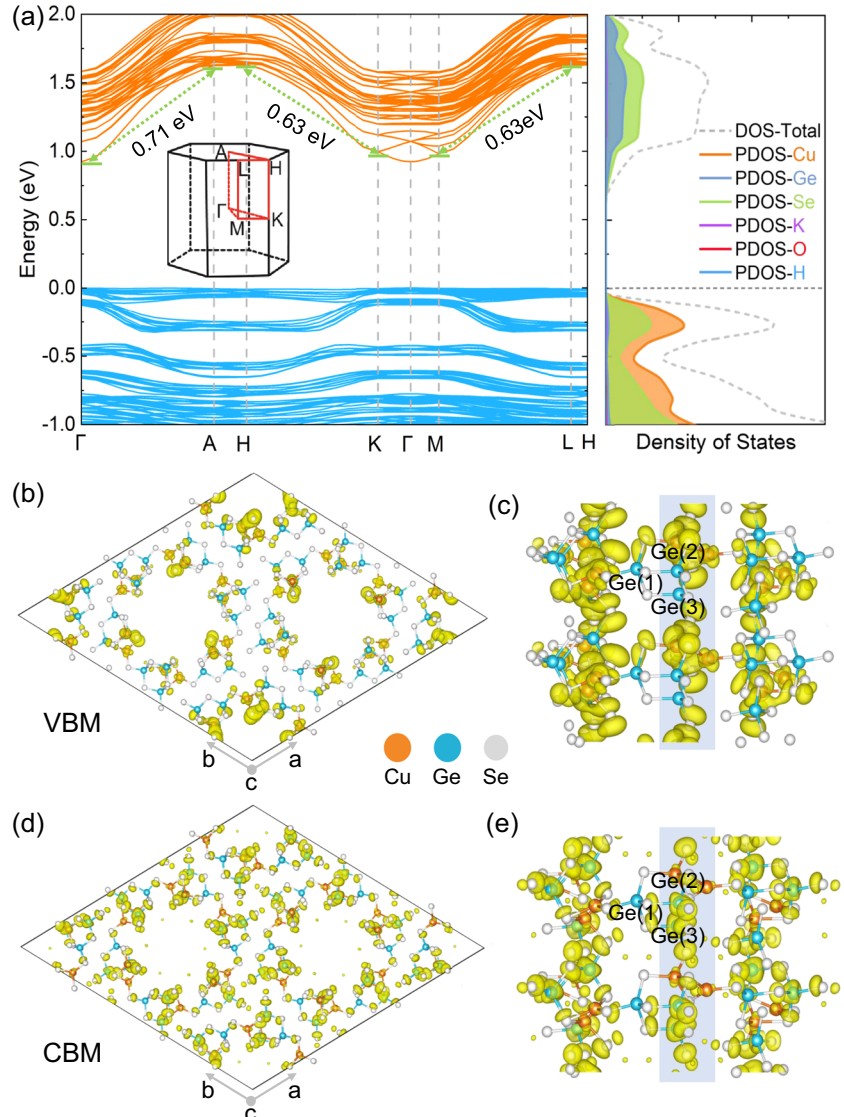

**Fig. 4 Theoretical investigation of electronic properties of 1. a** Left panel, calculated band structure; Right panel, density of states (DOS) and projected DOS (PDOS) of **1**, the gray dashed line at zero energy represents the Fermi level ($E_F$). **b**, **d** Front and (**c**), (**e**) side views of the charge density iso-surfaces (yellow region) summed over the bands near the VBM and the CBM of 1. For the side views of the VBM and the CBM, a small fragment of a single nanotube is selected for clarity. Colors: orange, Cu; blue, Ge; light gray, Se. As the K, O, and H atoms do not contribute to the electronic band edges, they are omitted for clarity.

**Details of the first-principle simulations**. All Our calculations were carried out using density functional theory (DFT) as implemented in the VASP program. The interaction between the core and valence electrons for all atoms in the system was described using the projector augmented wave (PAW) approach. The generalized gradient approximation (GGA) of the Perdew-Burke-Ernzerhof (PBE) functional was used for the exchange-correlation functional. Furthermore, in order to capture the weak van der Walls (VDW) interactions within this system, which was corrected by the Grimme DFT-D2 method. Notably, the crystal structure required for the simulation here is obtained by experimental analysis using single-crystal X-ray diffraction, and then detailed optimization of the atomic position based on the VASP software. The crystal structure was fully relaxed until total energies (atomic forces) converged to $10^{-4}$ eV (0.02 eV/Å) with the kinetic energy cutoff for plane-wave basis set to 400 eV. A $1 \times 1 \times 5$ Monkhorst–Pack k-point mesh has been used for structural optimization and a $2 \times 2 \times 10$ mesh has adopted for electronic structure calculation.

## Data availability

The X-ray crystallographic coordinates for 1 and 2 have been deposited at the Cambridge Crystallographic Data Center (CCDC), under deposition numbers CCDC 2052331 and 2052336. These data can be obtained free of charge from The Cambridge Crystallographic Data Center via www.ccdc.cam.ac.uk/data_request/cif. All remaining data are either providing in the Article and its supplementary information or available from the authors upon reasonable request. Source data are provided with this paper.

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

## Acknowledgements

This work was supported by the National Natural Science Foundation of China (21875150 and 22071165), and the 111 Project (D20015).

## Author contributions

T.W., J.T., and D.L. conceived the idea. T.W. and D.L. supervised the work. J.T., X.W., J.Z. prepared the samples and performed the corresponding characterizations. J.W. and W.Y. performed the DFT simulations. All authors contributed to the writing of the manuscript.

## Competing interests

The authors declare no competing interests.
