## [Peer Review File · Nature Communications]

REVIEWER COMMENTS

Reviewer #1 (Remarks to the Author):

The manuscript "An Unprecedented Chalcogenide cluster based Semiconducting Nanotube Array with Oriented Photoconductive Behavior" by Wu and coworkers presents the synthesis, the crystal structure and the electrical properties of an unprecedented tubular chalcogenide array based on supertetrahedral T₂-CuGeSe clusters. The 1D nanotubular structure of this compound is impressive since it presents a breakthrough in terms of the structure constructed by metal chalcogenide clusters. The described semiconducting nanotube array shows a fine electronic conductivity, the highest among reported crystalline nanotube arrays, and an oriented photoconductive behavior, which was confirmed by the related electrical experiments and theoretical calculations. This work not only enriches the kinds of crystalline inorganic nanotubes and but also provides an atomically precise structure model in many fields. As such, I would like to recommend its publication in Nature Communication providing the authors address the following concerns.

1. The structure of the cluster-based 1D nanotube is interesting. Is it possible to replace the metal or non-metal components by other elements of the same group with same valence? Or would it be possible to dope compound 1 with sulfur to modulate its electrical properties?
2. Is it possible to obtain individual nanotubes from dissociating the nanotube array of compound 1 like other crystalline nanotube arrays (J. Am. Chem. Soc. 2006, 128, 6538 and CCS Chemistry 2020, 2, 209)?
3. How about the chemical stability of compound 1 in air? Since Cu(I) compounds are easily oxidized in air.
4. There are also mistakes in grammar, for instance "... wheel-shaped ring was formed, which contrast to the 3D frameworks ...". And all the phrases of "compared to" should be amended to "compared with" or "in comparison to/with" according to the meaning of text.
5. For Figure S16 in the supporting information, the labels of synthesized and simulated are opposite to the real curves.
6. The conductivity of micro-crystals is tested with two-point probe method. Will it be better to conduct a 4-point probe measurement for this system?
7. How does the repeatability of the conductivity of compound 1 at room temperature? Is the reported value for the best result or an average of all results? The author should provide more information about the repeated experiments.
8. I notice that the conductivity of compound 1 is measured at 40 degrees Celsius, while the conductivity of compound 2 is measured at 25 degrees Celsius. I am worried about whether the conductivities of compound 1 and compound 2 could be compared, since they are measured at different temperature.
9. What does "a quasi-direct bandgap" mean? Is it an indirect bandgap or a direct bandgap? The author should explain it more.
10. What does the sentence "Moreover, the narrow band width and flat band lines near the VBM were attributed to the relatively larger localization of the Cu d orbital compared to the Ge s orbital." mean? According to the manuscript, the VBM of compound 1 is dominated by the Cu d orbital and the Se p orbital, so it is strange that the author referred to Ge s orbital in above sentence.
11. The author should discuss more about why the electric conductivity of compound 1 is much larger than that of compound 2.

Reviewer #2 (Remarks to the Author):

The major claim in this manuscript is the crystal structure of a crystalline nanotube array based on supertetrahedral chalcogenide clusters, and with a large internal diameter. AS previous crystalline nanotube arrays are based on oxides, the new material reported here has an improved electrical conductivity. To the best of my knowledge, this is indeed novel and merits publication. The new material reported here has been well characterised, with the characterisation data included in the

Supporting Information.

However, I disagree with the claim made by the authors at the end of the introduction, who state that "this is the first significant breakthrough since the emergence of the supertetrahedral Na₄Ge₄S₁₀T₂ cluster in 1971". Examples of significant breakthroughs in this field include:

- Report of 20 Å Supertetrahedral T₄ Clusters, J. Am. Chem. Soc. 2001, 123, 4867-4868
- Report of photoluminescence in supertetrahedra-based frameworks, Science 2002, 298, 2366-2369.
- Report of fast-ion conductivity in supertetrahedra-based frameworks, Nature 2003, 426, 428-432.

...and many other papers. Research activity in compounds containing supertetrahedral clusters has declined in recent years and the results presented here may open new opportunities in this field.

Other points:

1. I found the figures describing the crystal structure included in the Supporting Information clearer than Figure 1. In Figure 1, there isn't enough contrast between the grey atoms and the black background. Authors may want to change the colours in Figure 1 to improve clarity.
2. The description of the synthetic process should be improved. At the moment the manuscript states that, after the autoclave is open, red crystals are obtained by sonication. What is sonicated? The liquid contained inside the Teflon liner? Or is the solid product filtered? If solids are filtered, are they washed prior to sonication?
3. The yield given for compound 1 is "<1%". This is an extremely small yield, and will undoubtedly hinder exploitation of these results. Have the authors attempted to improve the yield?
4. The description of the electrical conductivity measurements is unclear. The statement "a direct current two terminal method were on KEITHLEY4200-SCS with an oven" should be rewritten. If measurements were performed on different crystals, these results should be included in the Supporting Information. Similarly, it seems that measurements as a function of temperature were carried out, and these should be included in the Supporting Information.

Overall, I feel this is an interesting manuscript that should be published, once my comments above have been addressed.

Reviewer #3 (Remarks to the Author):

This is an interesting manuscript, that reports a crystalline nanotube array based on a supertetrahedral chalcogenide cluster and exhibits an excellent structure-dependent electric conductivity and an oriented photoconductive behavior. I think it is of sufficient interest to the readers of Nature Communications, although I recommend a few revisions prior to publication.

1. Please provide the basis for the consistency of the 0.78nm interface lattice fringe observed by HRETM and the PXRD peak in compound 1 in the paper or supplementary materials, so that readers can better understand.
2. The summary of detailed structural information of inorganic and inorganic organic hybrid crystal nanotube arrays in Supplementary Table 2 needs to be supplemented. Some related Refs about crystalline inorganic nanotubes are missing. For example, the perfectly aligned 63 helical tubular cuprous bromide single crystal in Dalton Trans, 2015, 44, 3410. These Refs should be cited.
3. There are some minor errors need to be corrected in the paper, such as, "be0.52 ev" in the fourth line of the eighth paragraph.

Point-by-Point Responses to the Reviewers' Comments

Reviewer: 1

General Comments: *The manuscript “An Unprecedented Chalcogenide cluster based Semiconducting Nanotube Array with Oriented Photoconductive Behavior” by Wu and coworkers presents the synthesis, the crystal structure and the electrical properties of an unprecedented tubular chalcogenide array based on supertetrahedral T2-CuGeSe clusters. The 1D nanotubular structure of this compound is impressive since it presents a breakthrough in terms of the structure constructed by metal chalcogenide clusters. The described semiconducting nanotube array shows a fine electronic conductivity, the highest among reported crystalline nanotube arrays, and an oriented photoconductive behavior, which was confirmed by the related electrical experiments and theoretical calculations. This work not only enriches the kinds of crystalline inorganic nanotubes and but also provides an atomically precise structure model in many fields. As such, I would like to recommend its publication in Nature Communication providing the authors address the following concerns.*

General response: We really thank the reviewer for the valuable comments that are helpful to make the manuscript clearer and stronger. The reviewer’s comments are laid out and numbered below in *italicized font* while the detailed responses listed point-by-point are given in normal font and changes/additions to the manuscript are given in ***bold italic text*** and clearly highlighted in the revised manuscript. We sincerely hope that the following responses are helpful to clarify the reviewer’s concerns.

Specific Comments:

Comment 1: *The structure of the cluster-based 1D nanotube is interesting. Is it possible to replace the metal or non-metal components by other elements of the same group with same valence? Or would it be possible to dope compound 1 with sulfur to modulate its electrical properties?*

Response: Thanks very much for this constructive comment. In fact, we have tried to replace the metal or non-metal metal components of compound 1 by other elements of the same main group with same

valence to in the attempts of synthesis. Unfortunately, none of the attempts could produce the same structure as compound **1**. It is easy to synthesize a layered structure when metal Sn or salts of Sn were used to replace Ge element. When metal Ag or salts of Ag were used to replace Cu element, no products were obtained, and the reason for it may be the mismatch of atomic radii or the strong affinity between Ag and Se that leads to the formation of insoluble Ag₂Se instead of the similar tubular structures as compound **1**. Moreover, due to the rigidity of the M-Se bond, the tubular structure is already very special in the supertetrahedral cluster system and the co-doping of sulfur and selenium is more difficult to be achieved in this case, perhaps due to the mismatch of bond length and atomic radii, therefore, it is difficult to maintain the stability of the structure. Thanks again for the reviewer's insightful suggestion, and we hope that the above discussions would be satisfactory to clarify the reviewer's concerns.

Comment 2: *Is it possible to obtain individual nanotubes from dissociating the nanotube array of compound **1** like other crystalline nanotube arrays (J. Am. Chem. Soc. 2006, 128, 6538 and CCS Chemistry 2020, 2, 209)?*

Response: We are thankful for the reviewer's insightful suggestion. Considering compound **1** is constructed by packing of individual nanotubes, it is interesting to study the properties of an individual nanotube. In fact, we have tried to dissociate the nanotube array of compound **1** like other crystalline nanotube arrays by means of sonication in water and other common solvents or acid digestion with HNO₃ and other acids. However, due to the strong electrostatic interactions between negatively-charged nanotubes and positively-charged K⁺ ions that reside in the gap between the nanotubes (**Supplementary Figure 11**), our sonication attempts failed. While acid digestion would destroy the structure of compound **1** instead of separating individual nanotubes from the nanotube array as reported in the literature (*CCS Chemistry* **2020**, 2, 209). Thanks again for the reviewer's insightful comment, and we sincerely hope that the above discussions would be satisfactory to clarify the reviewer's concerns.

Comment 3: *How about the chemical stability of compound 1 in air? Since Cu(I) compounds are easily oxidized in air.*

Response: Thanks for this comment. As shown in Fig. R1, the structure of compound 1 could remain intact when stored in air for two days. After three days, the PXRD peaks became much weaker in comparison to the pristine sample, indicating the decomposition of the structure of compound 1 probably due to the easily oxidization by oxygen. Thus, compound 1 is usually stored in the glove box filled with N₂ for further use. This is mentioned and highlighted in the Methods part of revised manuscript (**Page 15, Lines 8-10**), please check it. We sincerely hope that the above discussions would be satisfactory to clarify the reviewer's concerns.

Fig. R1 PXRD patterns of as-synthesized compound 1 exposed in air for different periods.

Comment 4: *There are also mistakes in grammar, for instance “... wheel-shaped ring was formed, which contrast to the 3D frameworks” And all the phrases of “compared to” should be amended to “compared with” or “in comparison to/with” according to the meaning of text.*

Response: We sincerely thank the reviewer for the careful reading. In our revised manuscript, we have corrected those mistakes (“*which contrast to the 3D frameworks*” has been corrected into “*which is in*

sharp contrast to the 3D frameworks” (Page 7, Line 27); all “*compared to*” has been corrected into “*compared with*”) and all the corrections are clearly highlighted with yellow background in the revised manuscript, please check it.

Comment 5: *For Figure S16 in the supporting information, the labels of synthesized and simulated are opposite to the real curves.*

Response: Thanks very much for pointing out this error. The labels in Supplementary Figure 16 are amended (Page S16, Line 1), please check it.

Comment 6: *The conductivity of micro-crystals is tested with two-point probe method. Will it be better to conduct a 4-point probe measurement for this system?*

Response: Thanks for your constructive comments. The 4-point probe measurement is ideal for accurately measuring electrical conductivity. We have tried several methods to perform four-point probe measurement, including top-contact, bottom-contact and conductive paste methods (*J. Am. Chem. Soc.* **2016**, *138*, 14772), but all attempts failed. For the top-contact method, it is incompatible with our 25 μm thick single crystal because the evaporated metallic thin film is usually less than 1 μm thick. For the bottom-contact method, only one crystal surface could contact the electrodes due to the surface area of the contact point is relatively small, making it difficult to measure the cross-sectional area of the conducting channel. Moreover, as stated in the reference (*J. Am. Chem. Soc.* **2016**, *138*, 14772): “*Due to the small size of single crystals of MOFs (<1 mm), methods involving four contacts can be challenging.*”, we also encounter the similar situation. It is quite difficult to fabricate 4-contact onto a 1 mm-long crystal without short-circuiting the leads by means of using conductive paste method due to the small size and fragility of compound **1**. Meanwhile, the authors in the reference also pointed out that “*the typical resistance of wires and contacts is less than 100 Ω . Therefore, to measure electrical conductivity with less than 10% error, the resistance of the sample needs to be higher than 1 $\text{k}\Omega$* ”. Thus, the resistance that we measured should mainly be contributed by the compound **1** itself, since the resistance of compound **1** is about giga-ohm.

Comment 7: *How does the repeatability of the conductivity of compound 1 at room temperature? Is the reported value for the best result or an average of all results? The author should provide more information about the repeated experiments.*

Response: Thanks for your constructive comments. In fact, we have fabricated more than 10 devices and the reported value is not for the best result device and the result we reported is for one device that is close to the average value of 5 devices with a narrow range of conductivity. The results of 5 devices are provided in the Supplementary Information (**Supplementary Table 5**). From the experimental result, it could be seen that the measurement results of different devices are in a narrow range. The small difference in the electrical conductivity of these crystals may be attributed to the difference in contact resistance, crystal defects, and the fact that crystals obtained in different batches possess different thicknesses. As suggested by the reviewer, we have now revised the manuscript to make it clear. In the revised manuscript, the following sentences has been added to the main text: *“In addition, the repeatability of the conductivity of compound 1 is demonstrated on 5 individual devices, and the results showed that the electrical conductivities of 5 devices are in a narrow range (Supplementary Table 5).”* All related changes are highlighted with yellow background in the revised manuscript (**Page 11, Line 2-5**), please check it. Thanks again for the reviewer’s insightful suggestion and we hope that the above discussions would be satisfactory to clarify the reviewer’s concerns.

Comment 8: *I notice that the conductivity of compound 1 is measured at 40 degrees Celsius, while the conductivity of compound 2 is measured at 25 degrees Celsius. I am worried about whether the conductivities of compound 1 and compound 2 could be compared, since they are measured at different temperature.*

Response: We sincerely thank the reviewer for the careful reading. The conductivity of compound 1 at 25 °C is calculated to be $2.55 \times 10^{-6} \text{ S cm}^{-1}$ based on the good linearity of Arrhenius plots for compound 1 (**Fig. 2d**), which is much larger than that of compound 2 at 25 °C ($1.89 \times 10^{-9} \text{ S cm}^{-1}$). On the other hand, the conductivity of compound 2 measured at 40 °C is calculated to be $9.1 \times 10^{-9} \text{ S cm}^{-1}$ (**Supplementary Figure 21**), which is much smaller than that of compound 1 measured at 40 °C ($7.60 \times$

$10^{-6} \text{ S cm}^{-1}$). Therefore, we could see the conductivity of compound **2** is much smaller than that of compound **1** whether at 25 °C or 40 °C. To compare both conductivities at the same temperature conveniently, we have revised the conductivity of compound **2** in the main text: “*In the context of 2, the conductivity was determined to be only $9.1 \times 10^{-9} \text{ S cm}^{-1}$ at 40 °C*”. The revision is highlighted in the revised manuscript (**Page 11, Line 17-18**), please check it and we hope that it would be satisfactory to clarify the reviewer’s concerns.

Comment 9: *What does “a quasi-direct bandgap” mean? Is it an indirect bandgap or a direct bandgap? The author should explain it more.*

Response: Thanks for this insightful suggestion. For the indirect band gap semiconductor, as shown in the Fig. R2 below, when the difference ($\Delta E_g = E_{g\text{-indirect}} - E_{g\text{-direct}}$) between direct bandgap ($E_{g\text{-direct}}$) and indirect bandgap ($E_{g\text{-indirect}}$) is less than 0.02 eV, we call it "quasi-direct band gap" (*Chem. Mater.* **2017**, 29, 7868), because its electronic properties are basically the same as those of the direct bandgap at k_1 vector. Here, for the system of compound **1**, it could be regarded as a quasi-direct bandgap because the bandgap difference (ΔE_g) is about 0.01 eV. As suggested by the reviewer, we added some explanation and a reference in our revised manuscript. All revisions are highlighted (**Page 12, Lines 6-7**), please check it and we hope it would be satisfactory to clarify the reviewer’s concerns.

Fig. R2 Simple schematic illustration of the bandgap of indirect band gap semiconductors.

Comment 10: *What does the sentence “Moreover, the narrow band width and flat band lines near the VBM were attributed to the relatively larger localization of the Cu d orbital compared to the Ge s orbital.” mean? According to the manuscript, the VBM of compound 1 is dominated by the Cu d orbital and the Se p orbital, so it is strange that the author referred to Ge s orbital in above sentence.*

Response: Thanks for your insightful comments. According to the atomic-orbital projected band structures (**Supplementary Figure 23**), we can see that the upper valence bands are mainly contributed by Cu *d* orbital and Se *p* orbital. Since Cu *d* orbital here has a larger weight (represented by the size of solid circle, see **Supplementary Figure 23**) and is relatively localized, the width of upper valence band is narrow and not very dispersive, leading to the formation of flat band lines near the VBM. Regarding why we refer to the Ge *s* orbital here, we meant to compare with the lower conduction band, and show that the difference in their dispersion intensities comes from Ge *s* orbital near the CBM and Cu *d* orbital near the VBM. We hope that the above discussions would be satisfactory to clarify the reviewer’s concerns.

Comment 11: *The author should discuss more about why the electric conductivity of compound 1 is much larger than that of compound 2.*

Response: Thanks for your constructive comments. As compound 1 possesses more complex unidirectional connectivity than compound 2, it is helpful for the transport of electrons in compound 1. In addition, copper is much more conducive to electron transport than cadmium, which may also contributes to the much improved conductivity of compound 1 than compound 2. Above discussions have been added in the main text and highlighted (**Page 11, Lines 21-25**), please check it. We hope that would be satisfactory to clarify the reviewer’s concerns.

Reviewer: 2

General Comments: *The major claim in this manuscript is the crystal structure of a crystalline nanotube array based on supertetrahedral chalcogenide clusters, and with a large internal diameter. As previous crystalline nanotube arrays are based on oxides, the new material reported here has an improved electrical conductivity. To the best of my knowledge, this is indeed novel and merits publication. The new material reported here has been well characterized, with the characterization data included in the Supporting Information. However, I disagree with the claim made by the authors at the end of the introduction, who state that “this is the first significant breakthrough since the emergence of the supertetrahedral $\text{Na}_4\text{Ge}_4\text{S}_{10}$ T2 cluster in 1971”. Examples of significant breakthroughs in this field include:*

- Report of 20 Å Supertetrahedral T4 Clusters, J. Am. Chem. Soc. 2001, 123, 4867-4868

-Report of photoluminescence in supertetrahedra-based frameworks, Science 2002, 298, 2366-2369.

- Report of fast-ion conductivity in supertetrahedra-based frameworks, Nature 2003, 426, 428-432.

...and many other papers. Research activity in compounds containing supertetrahedral clusters has declined in recent years and the results presented here may open new opportunities in this field.

General response: We are grateful to the reviewer for the constructive comments and valuable suggestions that are helpful for us to make the manuscript clearer and stronger. And we totally agree with the reviewer’s opinion about the significant breakthroughs in compounds containing supertetrahedral clusters. We are very sorry that our pristine statement is not comprehensive and precise enough. Thus, we revised this sentence as “*...that it will constitute another significant breakthrough in the field of supertetrahedral clusters since the emergence of the supertetrahedral $\text{Na}_4\text{Ge}_4\text{S}_{10}$ T2 cluster...*” in our revised manuscript and the examples of significant breakthroughs in this field mentioned by the reviewer are all cited. All the revisions are highlighted (**Page 5, Lines 14-17**), please check it, and we sincerely hope that would be satisfactory to clarify the reviewer’s concerns. Moreover, the reviewer’s other comments are laid out and numbered below in *italicized font* while the detailed responses listed point-by-point are given in normal font and changes/additions to the manuscript are

given in ***bold italic text*** and clearly highlighted in the revised manuscript. We sincerely hope that the following responses are helpful to clarify the reviewer's concerns.

Specific Comments:

Comment 1: *I found the figures describing the crystal structure included in the Supporting Information clearer than Figure 1. In Figure 1, there isn't enough contrast between the grey atoms and the black background. Authors may want to change the colors in Figure 1 to improve clarity.*

Response: We highly appreciate the reviewer's constructive suggestion. As suggested by the reviewer, we have changed the color of grey atoms in Figure 1 to yellow to get a sharper contrast with the black background (**Page 6, Line 6**), please check it. And we sincerely hope that the modified Figure 1 would be satisfactory to clarify the reviewer's concerns.

Comment 2: *The description of the synthetic process should be improved. At the moment the manuscript states that, after the autoclave is open, red crystals are obtained by sonication. What is sonicated? The liquid contained inside the Teflon liner? Or is the solid product filtered? If solids are filtered, are they washed prior to sonication?*

Response: Thanks very much for the careful reading. The detailed synthetic process is described as follows: After cooled down to room temperature, all the products in the autoclave are transferred to a 10 mL glass vial, and the upper mother liquor is abandoned. Then *ca.* 2 ~ 4 mL ethylenediamine is added in the glass vial with sonication (40 kHz, 240 W) for *ca.* 1 min. After standing for 1 min, the upper liquid is abandoned. Repeat above steps with ethylenediamine until the upper liquid become clear with lighter color. Then repeat above steps with ethonal for 2 ~ 3 times to remove ethylenediamine. Finally, a small amount clean and pure red rod crystals could be obtained after filtration (yield: < 1%, based on Cu element). Above detailed process has been added to the Methods part and highlighted in the revised manuscript (**Page 14, Line 25 and Page 15, Lines 1-7**), please check it. And we sincerely hope that the above discussion would be satisfactory to clarify the reviewer's concerns.

Comment 3: *The yield given for compound 1 is “<1%”. This is an extremely small yield, and will undoubtedly hinder exploitation of these results. Have the authors attempted to improve the yield?*

Response: Thanks very much for the insightful comment. The reviewer is right, such small yield is indeed a difficult problem for further exploitation when we obtained compound **1** initially. After many attempts, we found that adding a small amount of trimesic acid (~10 mg) in the synthesis process could increase the crystal yield. But the specific reason is unknown at present. Besides, we used many autoclaves and carefully purified and collected compound **1** to obtain enough products for further exploitation. The adding of trimesic acid is mentioned in the Methods part of our revised manuscript (**Page 15, Lines 7-8**), please check it.

Comment 4: *The description of the electrical conductivity measurements is unclear. The statement “a direct current two terminal method were on KEITHLEY4200-SCS with an oven” should be rewritten. If measurements were performed on different crystals, these results should be included in the Supporting Information. Similarly, it seems that measurements as a function of temperature were carried out, and these should be included in the Supporting Information.*

Response: Thanks very much the constructive suggestion. According to the suggestion, we have revised the electrical conductivity measurement part in the Method of revised manuscript to make it clearer. The following description has been added to the electrical conductivity measurement part: “*Before preparing for electrical contact, the cylindrical crystals were covered by silver paste and connected to the semiconductor analysis system (4200SCS, Keithley) by gold wires on the insulating sapphire substrate. The lengths and widths of the column-shaped crystals on the substrate were estimated by means of microscope based on the width of the gold wire (diameter: 50 μm). Electrical conductivity (σ) is obtained by fitting the linear region of the current–voltage curves according to Ohm’s law. The activation energies reported herein were calculated by the Arrhenius Eq. (1) as $\ln k = \ln A - \frac{E_a}{RT}$ (1). The temperature-dependent I-V curve measurements for the single crystal of compound 1 and 2 were performed on KEITHLEY4200-SCS by means of a direct current two terminal method. The temperature is controlled by a digital automatic temperature control oven (STIK BAO-35A) and the*

measurements are performed in the range of 20-120 °C.” These revisions are clearly highlighted in Method part (**Page 15, Lines 19-26 and Page 16, Lines 1-6**), please check it. And we hope that the revised description of the electrical conductivity measurements would be satisfactory to clarify the reviewer’s concerns. As for the measurements of conductivity performed on different crystals, the results have been provided in the Supplementary Information (**Supplementary Table 5**). For the measurements of conductivity as a function of temperature, detailed corresponding description is provided and highlighted in the electrical conductivity measurement part as mentioned above (**Page 16, Lines 1-6**), please check it. Thanks for your suggestion again, and we sincerely hope that the above discussion would be satisfactory to clarify the reviewer’s concerns.

Reviewer: 3

General Comments: *This is an interesting manuscript, that reports a crystalline nanotube array based on a supertetrahedral chalcogenide cluster and exhibits an excellent structure-dependent electric conductivity and an oriented photoconductive behavior. I think it is of sufficient interest to the readers of Nature Communications, although I recommend a few revisions prior to publication.*

General response: We sincerely thank the reviewer for the insightful comments that are helpful for us to make the paper clearer and stronger. The reviewer's comments are laid out and numbered below in *italicized font* while the detailed responses listed point-by-point are given in normal font and changes/additions to the manuscript are given in ***bold italic text*** and clearly highlighted in the revised manuscript. We sincerely hope that the following responses are helpful to clarify the reviewer's concerns.

Specific Comments:

Comment 1: *Please provide the basis for the consistency of the 0.78 nm interface lattice fringe observed by HRETM and the PXRD peak in compound 1 in the paper or supplementary materials, so that readers can better understand.*

Response: We highly appreciate the reviewer's constructive suggestion. The consistency of the 0.78 nm interface lattice fringe observed by HRETM and the PXRD peak at $2\theta = 11.7^\circ$ in compound **1** is according to the Bragg's equation: $2 \times d \times \sin\theta = n \times \lambda$. Here, d is the interplanar spacing and also the distance of interface lattice fringe observed by HRETM, θ is a half of the diffraction angle of PXRD peak, λ is the wavelength of Cu-K α ($\lambda = 1.54184 \text{ \AA}$), n is the diffraction order ($n = 1$). The above basis is provided in the Supplementary Methods (**Page S2, Lines 13-15**), and relevant parts in the paper have also been modified accordingly (**Page 8, Lines 17-18**), please check it.

Comment 2: *The summary of detailed structural information of inorganic and inorganic organic hybrid crystal nanotube arrays in Supplementary Table 2 needs to be supplemented. Some related Refs about*

crystalline inorganic nanotubes are missing. For example, the perfectly aligned 63 helical tubular cuprous bromide single crystal in Dalton Trans, 2015, 44, 3410. These Refs should be cited.

Response: Thanks for this constructive suggestion. The recommended reference has been added as ref. 16 in the revised Supplementary Information, and relevant information has been supplemented in Supplementary Table 2. Please check it.

Comment 3: *There are some minor errors need to be corrected in the paper, such as, “be0.52 ev” in the fourth line of the eighth paragraph.*

Response: Thanks very much for the careful reading. We have carefully revised the manuscript again and corrected some minor errors, including the mentioned error (**Page 10, Line 10**). All revisions are highlighted with yellow background, please check it.

REVIEWERS' COMMENTS

Reviewer #1 (Remarks to the Author):

I think the authors have well addressed the comments from Reviewers, and the quality of the manuscript has been accordingly improved. Therefore, I would like to recommend the acceptance of the paper now.

Reviewer #3 (Remarks to the Author):

The manuscript reports a crystalline nanotube array based on supertetrahedral chalcogenide cluster, which is an important breakthrough in terms of the structure constructed by metal chalcogenide clusters. In addition, the nanotube exhibits an excellent structure-dependent electric conductivity and an oriented photoconductive behavior, which are well supported by related experiments and theoretical calculations. The author gave a good reply to the comments put forward by the reviewers, and made corresponding amendments in the manuscript to make the content more perfect. I strongly recommend its publication in *Nature Communication*.

Point-by-Point Responses to the Reviewers' Comments

Reviewer: 1

General Comments: *I think the authors have well addressed the comments from Reviewers, and the quality of the manuscript has been accordingly improved. Therefore, I would like to recommend the acceptance of the paper now.*

General response: We really thank the reviewer for the positive comments.

Reviewer: 3

General Comments: *The manuscript reports a crystalline nanotube array based on supertetrahedral chalcogenide cluster, which an important breakthrough in terms of the structure constructed by metal chalcogenide clusters. In addition, the nanotube exhibits an excellent structure-dependent electric conductivity and an oriented photoconductive behavior, which are well supported by related experiments and theoretical calculations. The author gave a good reply to the comments put forward by the reviewers, and made corresponding amendments in the manuscript to make the content more perfect. I strongly recommend its publication in Nature Communication.*

General response: We sincerely thank the reviewer for the positive comments.